# Gender-Specific Prognostic Impact of Treosulfan Levels in High-Dose Chemotherapy for Multiple Myeloma

**DOI:** 10.3390/cancers16193364

**Published:** 2024-10-01

**Authors:** Alexander D. Heini, Karin Kammermann, Ulrike Bacher, Barbara Jeker, Michael Hayoz, Yolanda Aebi, Carlo R. Largiadèr, Henning Nilius, Thomas Pabst

**Affiliations:** 1Department of Medical Oncology, University Hospital Inselspital and University of Bern, 3010 Bern, Switzerland; 2Department of Hematology and Central Hematology Laboratory, University Hospital Inselspital and University of Bern, 3010 Bern, Switzerland; 3Center of Laboratory Medicine (ZLM), University Hospital Inselspital and University of Bern, 3010 Bern, Switzerland; 4Department of Clinical Chemistry, University Hospital Inselspital and University of Bern, 3010 Bern, Switzerland; 5Graduate School for Health Sciences, University of Bern, 3012 Bern, Switzerland

**Keywords:** treosulfan, pharmacokinetics, gender difference, myeloma, high-dose chemotherapy (HDCT), autologous stem cell transplantation (ASCT)

## Abstract

**Simple Summary:**

High-dose chemotherapy (HDCT) with autologous stem cell transplantation (ASCT) is a standard treatment for multiple myeloma (MM). This study examines gender-specific differences in response to treosulfan and melphalan (TreoMel) HDCT. Data from 112 MM patients treated at a single center were analyzed for outcomes such as response rate, progression-free survival (PFS), overall survival (OS), and toxicities. Results revealed that females had significantly higher treosulfan exposure than males, as shown by peak levels and area under the curve (AUC). Notably, higher treosulfan exposure in females was associated with increased mortality, with those having an AUC > 900 mg*h/L showing shorter OS. However, treosulfan exposure did not affect PFS in females, and no significantly increased mortality risk was observed in males. These findings suggest that higher treosulfan doses increase toxicity in females without improving outcomes, supporting the need for further investigation into lower dosing for female MM patients.

**Abstract:**

Introduction: The growing body of evidence around sexual and gender dimorphism in medicine, particularly in oncology, has highlighted differences in treatment response, outcomes, and side effects between males and females. Differences in drug metabolism, distribution, and elimination, influenced by factors like body composition and enzyme expression, contribute to these variations. Methods: We retrospectively analyzed data of 112 multiple myeloma (MM) patients treated with first-line high-dose chemotherapy (HDCT) with treosulfan and melphalan (TreoMel) followed by autologous stem cell transplantation (ASCT) at a single academic center between January 2020 and August 2022. We assessed response rate, progression-free survival (PFS), overall survival (OS), and toxicities in relation to gender and treosulfan exposure. Results: Our analysis revealed significant gender-specific differences in treosulfan exposure. Females had higher peak levels (343.8 vs. 309.0 mg/L, *p* = 0.0011) and area under the curve (AUC) (869.9 vs. 830.5 mg*h/L, *p* = 0.0427) compared to males. Higher treosulfan exposure was associated with increased mortality in females but not in males. Females with treosulfan AUC > 900 mg*h/L had significantly shorter overall survival, while PFS was unaffected by treosulfan exposure. Conclusion: Our study demonstrates that female patients undergoing TreoMel HDCT have higher treosulfan exposure than males and that females with higher levels are at increased risk for toxicity and adverse outcomes. These data suggest that higher treosulfan doses do not confer a benefit in terms of better outcomes for females. Therefore, exploring lower treosulfan doses for female MM patients undergoing TreoMel HDCT may be warranted to mitigate toxicity and improve outcomes.

## 1. Introduction

The concept of sex and gender dimorphism has gained rapidly growing attention in medicine. Dimorphism refers to differences in biology between non-sex-related cancers arising in females and males, which can affect response to treatment, outcome, and side effects [1]. Many factors affecting drug distribution and elimination show marked differences between females and males. Even when adapting for weight and size, differences in body composition with greater plasma volume, organ perfusion, and higher body fat percentage in females lead to longer drug elimination time and increased drug concentration. Lower glomerular filtration with renal function being roughly 20% lower in females leads to decreased elimination of unchanged drugs [2]. Variations in expression of enzymes involved in drug metabolism, such as cytochrome P450, lead to slower elimination of active compounds [3]. These factors combined lead to increased exposure of female patients to some drugs.

Several reports on gender differences in elimination of anticancer agents have been published. These include cytotoxic agents such as 5-fluorouracil [4,5] and doxorubicin [6], monoclonal antibodies [7], as well as targeted agents including tyrosine kinase inhibitors such as imatinib [8]. These differences can have important clinical implications. Unger et al. analyzed the data of more than 23,000 patients involved in phase II and III trials and concluded that females were more likely to suffer from severe treatment-related adverse events (SAE) than males (odds ratio (OR) 1.34). The gender differences in SAE were significant irrespective of the type of AE and the type of therapy (chemotherapy, antibody, or targeted agent) [9].

In hematologic oncology, a small number of studies have investigated differences in response and prognosis of males and females. Males have been shown to be associated with adverse survival in elderly patients with DLBCL undergoing R-CHOP chemotherapy [7], since elderly males have a higher clearance of rituximab than females, resulting in decreased rituximab exposure. The findings of the SEXIE-R-CHOP trial demonstrated that increasing the dose of rituximab to 500 mg/m^2^ in male patients (from the conventional 375 mg/m^2^) contributed to overcoming the adverse prognosis as it approximated the outcomes of male patients to those of the female study participants [10]. Similarly, in the phase III ECHELON-2 study, which investigated brentuximab vedotin (BV) in patients with peripheral T cell lymphoma, subgroup analyses suggested that female patients derived a greater benefit from BV compared to male patients [11].

Multiple myeloma (MM) is a plasma cell disorder and is the second most common hematologic malignancy in adults. Males are affected more frequently, with the age-standardized incidence rate being about 1.6 times higher than in females [12]. In an analysis of close to 4000 patients enrolled in the Myeloma XI trial, Bird et al. reported significant differences in baseline hemoglobin and platelet counts as well as kidney function between females and males. Moreover, some genetic lesions were observed more frequently in females, such as t(14;16), associated with adverse prognosis. Similar results were also observed in the earlier Myeloma IX trial [13].

Despite these differences in baseline characteristics, the impact of gender on response and outcome in MM patients is less clear. While the Myeloma IX trial showed inferior OS in female patients (median 44.8 months vs. 49.9 months), Myeloma XI data indicated no survival differences. Other investigations showed either no differences in outcome between male and female patients [14] or a trend towards adverse survival in male patients [15]. Gender therefore does not seem to be an independent prognostic factor in MM.

High-dose chemotherapy (HDCT) with autologous stem cell transplantation (ASCT) is a key component in consolidation therapy for multiple myeloma, especially for younger and middle-aged patients, as it provides durable remissions with prolonged intervals until next treatment. However, patient selection is crucial, as side effects can be considerable. Since three decades, high-dose melphalan at a dose of 200 mg/m^2^ is considered the standard of care [16]. Randomized trials have shown a marked improvement in PFS; however, so far, no OS benefit has been demonstrated [17,18]. The addition of compounds such as busulfan, BCNU, or bendamustine to melphalan has improved response rates in some studies, but no OS benefits have so far been identified using this strategy [16,19].

Subgroup analyses of the trials above have shown no differences in outcome between male and female patients. However, prospective data on gender-specific differences in outcome after HDCT/ASCT are lacking. Posch et al. retrospectively analyzed data of 191 patients with MM undergoing HDCT/ASCT and found no significant differences in outcome between male and female patients [14]. Also, in the Griffin trial [20], which investigated the addition of daratumumab to induction therapy with lenalidomide, bortezomib, and dexamethasone, no differences in outcome depending on gender were found.

Our group recently reported on the addition of treosulfan to melphalan (TreoMel) for HDCT in MM patients [21]. In this prospective trial, we found promising activity with a CR rate of 84% and a manageable safety profile. Moreover, we found female patients to have increased exposure to treosulfan compared to male patients with higher peak levels and area under the curve (AUC).

In this present study, we investigated the gender-specific impact of elevated treosulfan levels on the outcomes of MM patients who were treated with TreoMel HDCT in first remission. To our knowledge, gender-specific aspects in this treatment setting were so far not separately investigated.

## 2. Materials and Methods

### 2.1. Study Design and Patient Cohort

We performed a retrospective analysis of MM patients undergoing HDCT/ASCT consolidation with TreoMel in first remission. We assessed response rate, progression-free survival (PFS), and overall survival (OS), as well as toxicities, and results were correlated with gender and with treosulfan pharmacokinetics. The consecutive patients included in the study were treated between January 2020 and August 2022 at a single academic institution (Inselspital, University Hospital of Bern, Bern, Switzerland). All patients gave written informed consent for the use of their data. The study was approved by the local ethics committee of Bern, Switzerland (Ethics Commission of the Canton of Bern, decision number 2024-00927) and performed in accordance with the Declaration of Helsinki.

### 2.2. Stem Cell Mobilization, Apheresis and TreoMel HDCT Treatment Schedule

For the preceding stem cell mobilization, most patients were treated in the MOCCCA trial (NCT NCT03442673) and received either G-CSF alone or chemotherapy with either gemcitabine or vinorelbine in combination with G-CSF. After stem cell apheresis, high-dose chemotherapy treatment was administered as previously described [21]. Briefly, all patients were given treosulfan at 14 g/m^2^ i.v. on days −4, −3, and −2. Melphalan was administered on day −1 at 200 mg/m^2^ i.v. Anti-emetic and anti-infective prophylaxis with trimethoprim–sulfamethoxazole, fluconazole, and valacyclovir was given according to local institutional guidelines [21].

Stem cells were administered on day 0 after premedication with methylprednisolone and clemastine. To ensure timely neutrophil recovery, 5 μg/kg/day of filgrastim was administered between days +6 and +12. To prevent engraftment syndrome, dexamethasone was administered on days +9 to +13. All patients were given allopurinol during HDCT and prior to stem cell infusion, and following stem cell infusion, zoledronic acid was administered at day +1 and folic acid daily starting from day +1 and up to 8 weeks.

### 2.3. Assessment of Treosulfan Pharmacokinetics

Treosulfan pharmacokinetics were assessed as previously described [21]. In summary, treosulfan levels in peripheral blood samples were assessed on day −3 of the HDCT schedule (thus, at the second day of treosulfan administration to ensure steady-state conditions). After an initial sample was collected before the start of the treosulfan infusion, five further samples were taken 30, 60, 120, 240, and 360 min after completion of the infusion. Blood samples were stabilized by adding a sodium citrate buffer and stored at −80 °C until analysis immediately after collection.

As previously described [22], treosulfan concentrations were assessed by ultra-performance liquid chromatography tandem mass spectrometry (UPLC-MS/MS). Measurements were obtained through multiple reaction monitoring (MRM) using a Xevo TQ-S mass spectrometer (Waters Corp., Milford, MA, USA), operating in a positive ion electrospray ionization mode. Data was processed in TargetLynx (MassLynx software, version 4.1, Waters Corp., Milford, MA, USA) by integration of the area under the specific multiple reactions monitoring chromatograms compared to the area of the isotope-labeled analog.

Treosulfan area under the curve was calculated using Microsoft Excel (Microsoft Office version 16.8, Microsoft Corporation, Redmond, WA, USA) using the trapezoidal rule.

### 2.4. Response Assessment and Outcome Measures

Initial staging was performed according to the revised international staging system (R-ISS) [23]. Patients were classified as high-risk cytogenetics by the detection of at least one of the following genomic alterations: t(4;14), t(14;16), or del(17p). Response assessment was performed according to the International Myeloma Working Group (IMWG) criteria.

Progression-free survival was calculated from ASCT (day 0) to disease progression or death from any cause, whichever occurred first, and patients were censored at their last follow-up. Overall survival was calculated from ASCT (day 0) to death from any cause, and patients alive were censored at their last follow-up.

### 2.5. Statistical Analysis

Data cutoff was on 31 October 2023. Data curation was performed in Microsoft Excel Version 16 (Microsoft Corporation, Redmond, WA, USA). Statistical and survival analyses were performed in GraphPad Prism Version 9 (GraphPad Software, Boston, MA, USA). All *p* values stem from unpaired *t* tests or Pearson’s chi square tests if not stated otherwise. Survival was calculated using the Kaplan–Meier method and compared using log–rank tests.

Multivariable analyses were conducted in R (Version 4.0.2, R Foundation for Statistical Computing, Vienna, Austria) by fitting a Cox proportional hazards model to the data. The multivariable model for progression-free survival was adjusted for age, gender, R-ISS, cytogenetic risk group, remission status before HDCT/ASCT, treosulfan peak, and AUC. The multivariable model for overall survival was adjusted for all of the above predictors except R-ISS, because no deaths occurred at the R-ISS stage I in the study population.

## 3. Results

### 3.1. Baseline Characteristics

We identified 112 consecutive patients treated with TreoMel high-dose chemotherapy in first remission at our institution in the study period. Baseline characteristics are summarized in Table 1. Characteristics were balanced between female and male patients. However, male patients were younger than females (median age at diagnosis 60.8 vs. 64.3 years, *p* = 0.0413). We identified no differences in kidney function, and MM-specific disease characteristics such as R-ISS stages or high-risk cytogenetics were equally distributed.

### 3.2. Treatment before HDCT/ASCT

The initial therapy is summarized in Table 2. Most patients were treated with bortezomib, lenalidomide, and dexamethasone (VRd); however, female patients were more likely to have received daratumumab, most of them in combination with VRd: 23.8% of females were treated with daratumumab during induction and only 5.7% of males (*p* = 0.0075). While the median number of cycles of induction treatment was the same for female and male patients, time to HDCT/ASCT was slightly longer in females (5.3 vs. 4.6 months, *p* = 0.0056). Assessment after induction therapy showed tended towards improved response in females (VGPR or better in 71.4% vs. 52.9%, *p* = 0.0729), possibly affected by the more frequent use of daratumumab in female MM patients in our cohort.

### 3.3. HDCT/ASCT Therapy

All patients underwent HDCT with TreoMel at the planned dose. The median number of transfused CD34+ cells was comparable in both females and males.

### 3.4. Treosulfan Pharmacodynamics

Treosulfan is a prodrug that is converted non-enzymatically, in a pH-dependent manner, to its active metabolite, L-diepoxybutane. The active compound alkylates DNA at guanine residues to form crosslinks, resulting in DNA fragmentation and apoptosis. Around 15–40% of an administered dose of treosulfan are excreted unchanged in the urine, while the active compounds are eliminated after undergoing metabolism mainly in the liver and lung [24]. Due to the high clearance of 150–300 mL/min, the half-life of treosulfan is short, estimated at 1.5–2 h [25].

Pharmacodynamics are summarized in Figure 1A,B. In our cohort, we found treosulfan levels to be significantly higher in females than in males (median peak levels 343.8 vs. 309.0 mg/L, *p* = 0.0011, median AUC 869.9 vs. 830.5 mg*h/L, *p* = 0.0427). Peak treosulfan levels were reached within 30 min after the end of the infusion, and concentrations declined steadily afterwards.

### 3.5. Outcome

We analyzed survival according to gender, and survival data are given in Figure 1C,D, and Table 3. After a median follow-up of 31 months, we documented no significant differences in survival between the males and females; median progression-free survival was 36.7 months in males and not reached in females (*p* = 0.2644), and overall survival was not reached in females or males (*p* = 0.8951). At 12 months, estimated OS rates were 91.8% for females and 96.9% for males (*p* = 0.1209), and PFS rates were 91.2% and 89.5%, respectively (*p* = 0.8094). In the multivariable analysis, gender was not associated with outcome. As expected, we found responses to deepen after ASCT—before ASCT, 59.9% of patients achieved a VGPR or better, and this number increased to 91.9% after ASCT with no significant differences between males and females.

### 3.6. Outcome According to Treosulfan Exposure

We next analyzed survival according to treosulfan exposure as measured by AUC and peak level. Results are summarized in Figure 2. Patients with treosulfan AUC above the median (833 mg*h/L) showed no difference in survival to patients with AUC below the median (*p* = 0.5043 for PFS and 0.5443 for OS, Figure 2A,B). Similarly, no difference in survival was found between patients with treosulfan peak levels above the median (330 mg/L) compared to patients with peak levels below the median (*p* = 0.4723 for PFS and 0.5322 for OS, Figure 2C,D).

Upon elevation of the AUC cutoff to 900 mg*h/L, we found a trend towards adverse PFS (*p* = 0.1900, Figure 2E) and significantly adverse OS (*p* = 0.0382, Figure 2F) in patients with higher AUC levels. Elevation of the peak cutoff to 400 mg/L also led to a trend towards adverse PFS and significantly adverse OS (*p* = 0.1885 and 0.0223, respectively, Figure 2G,H).

In the multivariable analysis, high treosulfan peak levels (>400 mg/L) remained a predictor for significantly adverse overall survival (HR 33.0 (1.65–658, *p* = 0.0220), Appendix A) while for PFS, the boundary for statistical significance was narrowly missed (HR 4.57 (0.96–21.9), *p* = 0.0570) as was for treosulfan AUC (HR for OS 2.89 (0.41–20.5, *p* = 0.3000)). High cytogenetic risk was the single most important predictor of outcome (HR for OS 87.3 (3.94–1935), *p* = 0.0050). Also, patients who did not achieve at least a partial response to induction therapy showed considerably adverse outcome (HR for OS 52.6 (1.09–2543, *p* = 0.0450).

Finally, we analyzed the impact of treosulfan levels on survival in males and females separately. Females with higher AUC were at risk for significantly adverse outcome (HR for death 10.72 (1.084–106.1, *p* = 0.0424), Figure 3A,B), while for PFS, the boundary for statistical significance was narrowly missed (HR for death or progression 5.374, 0.9134–31.6). Also, for peak level, significance was narrowly missed (Figure 3E,F). For males, no significant differences in survival were found (Figure 3C,D,G,H).

### 3.7. Toxicity

Toxicities related to TreoMel HDCT are summarized in Table 4. We found female patients to have shorter neutrophil recovery time (median 11 vs. 12 days, *p* = 0.0119); however, females were more likely to require red blood cell transfusions (median 0 vs. 1 unit transfused, *p* = 0.0173). Time to platelet recovery and number of units of platelets transfused did not differ between female and male patients.

Hospitalization duration was longer in females (median 24 vs. 21 days, *p* = 0.0007), and the rate of ICU admission was slightly higher in females, without statistical significance (11.9 vs. 2.9%, *p* = 0.1007).

Neutropenic fever, which is a common side effect after high-dose chemotherapy treatment, occurred in 97.6% of females and 97.1% of males, *p* > 0.9999. The most frequent site of infection was neutropenic colitis in 66.7% of females and 80% of males (*p* = 0.1223). Males were more likely to develop central line-associated blood stream infections (CLABSI) at 25.7% vs. 7.1% without statistical significance (*p* = 0.0732).

Females were slightly more likely to require parenteral nutrition (95.2% vs. 85.7%, *p* = 0.2051), and the duration of PN was longer in females than in males (median 10 vs. 8 days, *p* = 0.0025). Mucositis was not different between females and males (61.7% vs. 47.5%, *p* = 0.5585), but refeeding syndrome was more likely to occur in males (55% vs. 32.5% of patients requiring PN, *p* = 0.0401).

Other side effects of interest were acute kidney injury, which occurred in 9.5% of female and 4.3% of male patients (*p* = 0.4218), and cardiac insufficiency, which occurred more frequently in female than in male patients (11.9% vs. 1.4%, *p* = 0.0271).

## 4. Discussion

Recent advances in understanding gender dimorphism in medicine and in hematologic oncology in particular underscore the importance of investigating variations in disease incidence, treatment responses, and outcomes between male and female patients [10,11], Understanding the underlying mechanisms driving these differences is the key to developing tailored treatment strategies that can improve outcomes for both females and males—or decrease side effects. By integrating these insights into clinical practice, there is potential to enhance the personalization of care in hematologic oncology. Despite growing interest and research in this area, significant changes in clinical practice have been limited. Our current study contributes to this evolving field by highlighting gender-specific pharmacokinetics and outcome in patients with MM undergoing TreoMel HDCT/ASCT.

In this current study including over 110 consecutive patients, we confirm our recent finding [21] that female MM patients exhibit higher treosulfan peak levels and AUC than their male counterpart. Higher treosulfan exposure was associated with increased mortality and adverse survival outcomes, particularly among females. This suggests that while females experience higher treosulfan exposure, it does not translate into improved efficacy but rather correlates with increased toxicity and adverse outcomes.

Specifically, female patients with treosulfan AUC > 900 mg*h/L had significantly shorter OS; the significance for peak levels > 400 mg/L was narrowly missed. In contrast, higher treosulfan exposure did not adversely affect outcome in male patients; however, it has to be noted that patient numbers in the respective subgroup were small. In a multivariable analysis, higher treosulfan peak plasma levels remained an independent predictor of adverse overall survival for all patients. Importantly, our study found no significant impact of elevated treosulfan levels on PFS, indicating that higher exposure does not enhance the anti-myeloma effect. Conversely, Pai et al. reported on pediatric patients with thalassemia major who underwent treosulfan-based conditioning for allogeneic transplantation and found lower treosulfan exposure to be predictive of inferior 1-year thalassemia-free survival [26]. While the cohort was similar in size to our study population, valid comparison is challenging due to key differences, most notably the distinct context of allogeneic transplantation.

At an AUC of 858 ± 184.3 mg*h/L, treosulfan exposure in our cohort was comparable to previous reports by our group [22]; however, it was lower than described in the literature, probably due to differences in patient population. Reasons for the high interpatient variability have been described previously [22,27].

High treosulfan exposure has been found to be associated with increased risk for side effects. Van der Stoep et al. found increased rates of skin (OR 4.51) and mucosal (OR 4.40) toxicity in pediatric patients with higher treosulfan exposure; however, they did not find an association with outcome [28]. In our cohort, we found female patients to be at greater risk for toxicity, with mainly gastrointestinal side effects being more frequent. They were more likely to require parenteral nutrition and were on parenteral nutrition for longer than male patients (10 vs. 8 days), indicative of higher rates of treosulfan-mediated mucositis. In a previous analysis by our group investigating treosulfan-based HDCT/ASCT in patients with acute myeloid leukemia (AML), treosulfan exposure correlated with diarrhea severity in females older than 55 years [29]. Moreover, a longer inpatient stay is indicative of a greater amount of time required for recovery.

Interestingly, in the multivariable analysis of the whole cohort, only treosulfan peak levels, not AUC, remained predictive of survival. This could indicate that high peak levels are responsible for adverse outcomes rather than total drug exposure. A similar observation was made for bortezomib, a backbone of multiple myeloma induction therapy, where, compared to intravenous administration, subcutaneous application achieves similar exposure but decreased peak levels, reducing the incidence of neurotoxicity [30].

Strengths of our investigation are its large, homogeneous patient cohort consisting of more than 110 consecutive patients treated in first remission within 2.5 years, with minimal missing baseline data. Age and gender distribution were within the expected range. Limitations are the retrospective design limiting statistical validity. As data were analyzed from chart review and no grading of adverse events was initially performed, mild adverse events may have been underreported, and the interpretation of toxicities may require cautiousness. This is particularly true for mucositis, which is a common side effect of treosulfan, for which mild symptoms might not have been explicitly recorded. This is supported by the more frequent need for and the longer duration of parenteral nutrition in females.

In the randomized phase II TreoMel trial (NCT05636787) currently recruiting at our center, we will prospectively investigate the impact of the addition of treosulfan to melphalan on outcome and toxicities in MM patients undergoing HDCT/ASCT in first remission. This trial will allow us to investigate treosulfan pharmacokinetics and pharmacodynamics as well as toxicities in a prospective manner.

## 5. Conclusions

In conclusion, our study adds to the growing body of evidence underscoring the importance of investigating gender differences in clinical outcomes, which may have significant implications for patient care. In our patient cohort, females experienced higher treosulfan exposure, which correlated with increased treatment-related side effects. Importantly, females with elevated treosulfan levels were at a greater risk of adverse outcomes, whereas male patients appeared less affected by higher treosulfan exposure.

While this current study does not conclusively recommend dose adjustments for female patients, it underscores the need for further research to explore optimal dosing strategies and minimize toxicity.

## Figures and Tables

**Figure 1 cancers-16-03364-f001:**
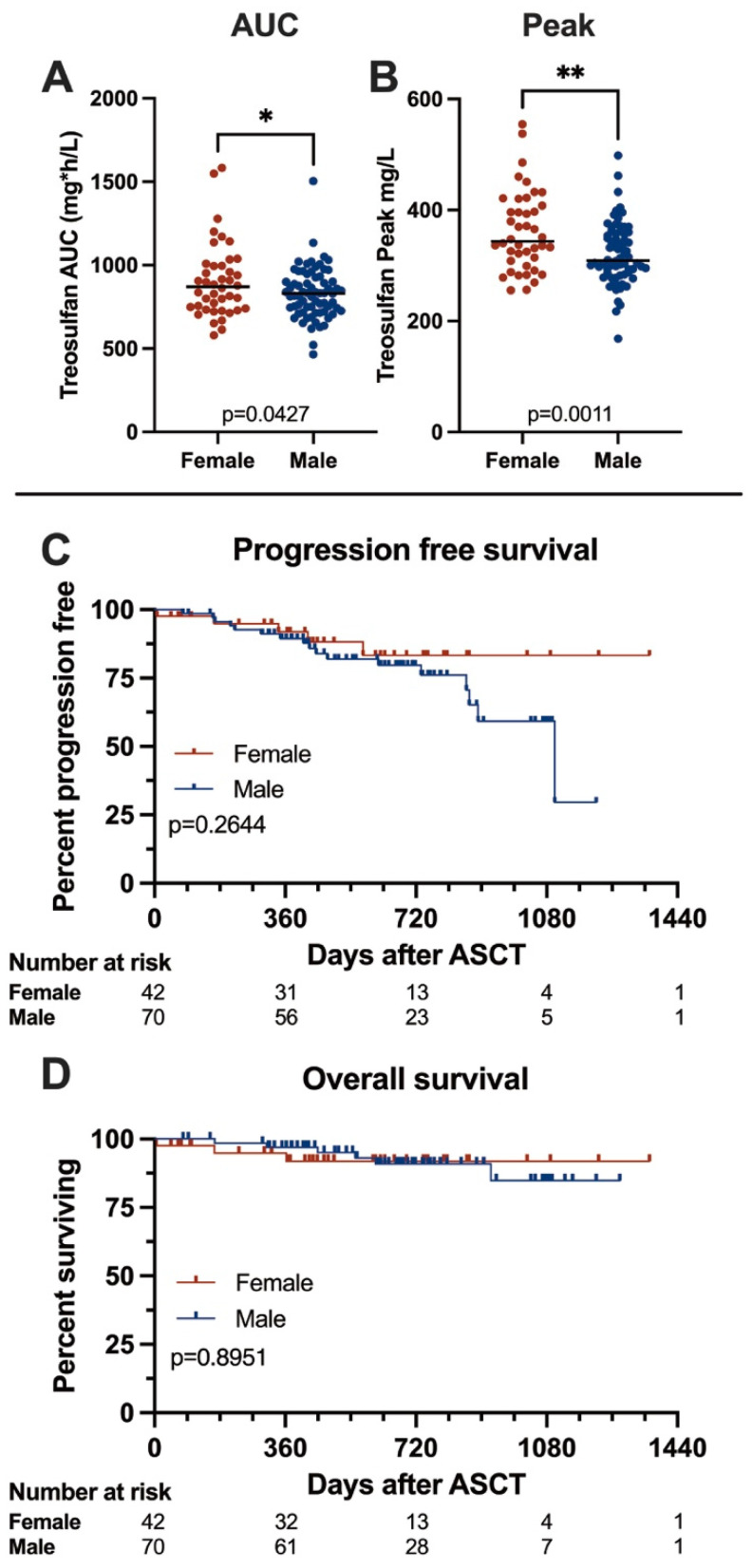
Treosulfan pharmacokinetics and Kaplan–Meier survival curves comparing female and male patients. (**A**,**B**) Comparison of treosulfan AUC and peak levels in female and male patients. Females showed significantly higher treosulfan exposure than males. (**C**,**D**) Progression-free and overall survival in female and male patients. We found no significant differences between genders. Asterisks denote significance level: n.s. *p* > 0.05; * *p* ≤ 0.05; ** *p* ≤ 0.01.

**Figure 2 cancers-16-03364-f002:**
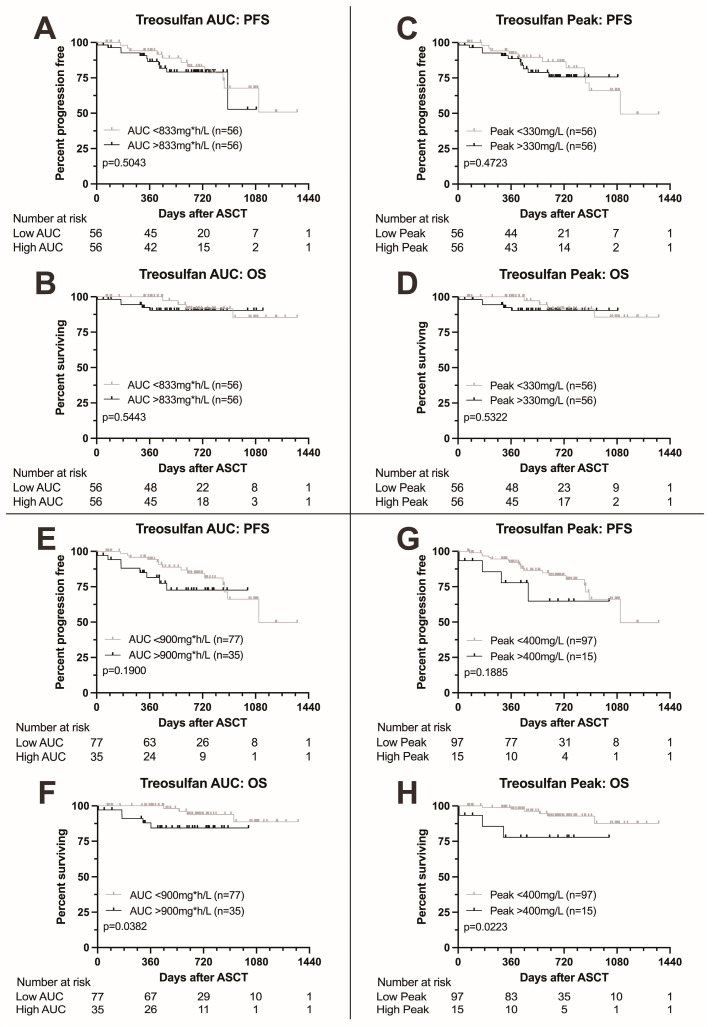
Kaplan–Meier survival curves according to treosulfan exposure. (**A**,**B**) Progression-free and overall survival according to treosulfan AUC above or below the median; no difference in progression-free or overall survival. (**C**,**D**) Progression-free and overall survival according to treosulfan peak level above or below the median; no difference in progression-free or overall survival. (**E**,**F**) Progression-free and overall survival according to treosulfan AUC above or below 900 mg*h/L. Higher treosulfan AUC was associated with adverse overall survival. (**G**,**H**) Progression-free and overall survival according to treosulfan peak level above or below 400 mg/L. Higher treosulfan peak levels were associated with adverse overall survival.

**Figure 3 cancers-16-03364-f003:**
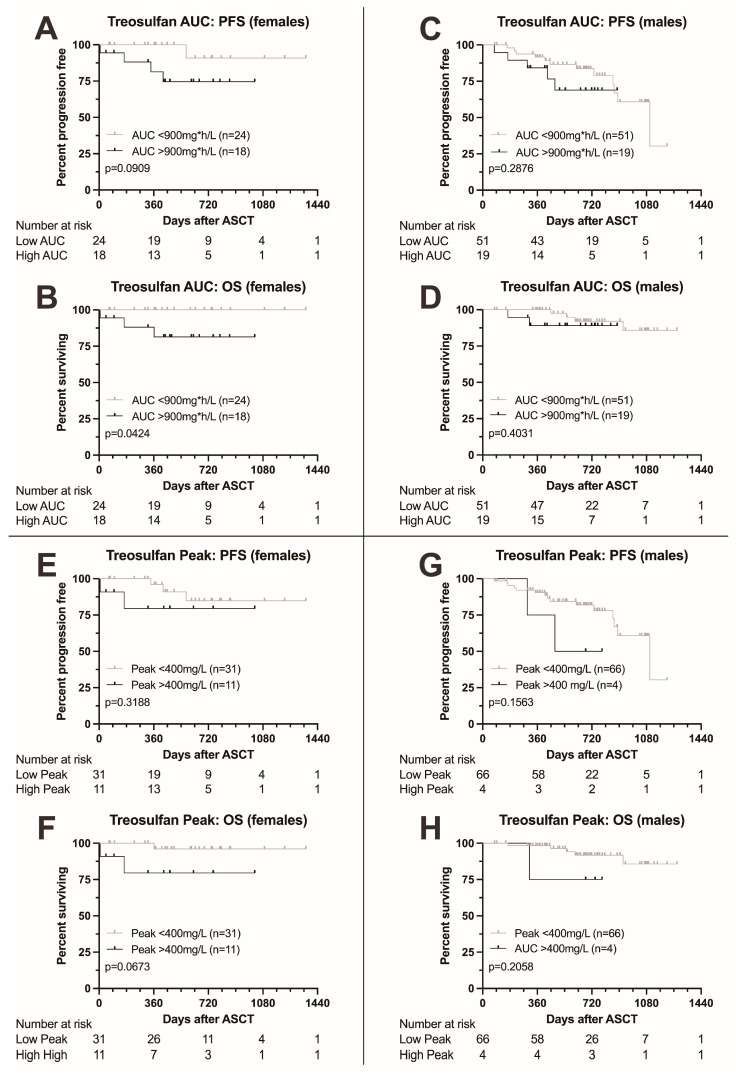
Kaplan–Meier survival curves according to treosulfan exposure, comparison between female and male patients. (**A**–**D**) Progression-free and overall survival in female (**A**,**B**) and male patients (**C**,**D**) according to treosulfan AUC (above or below 900 mg*h/L). OS was significantly lower in female patients with a higher AUC; males showed no difference. Moreover, a trend towards adverse PFS was found in female patients. (**E**–**H**) Progression-free and overall survival in female (**E**,**F**) and male patients (**G**,**H**) according to treosulfan peak levels (above or below 400 mg/L). A trend towards adverse overall survival was documented in females (**F**), with no significant differences observed in males.

**Table 1 cancers-16-03364-t001:** Baseline characteristics of the patients that participated in this study.

	Females	%/Range	Males	%/Range	*p*-Value	s	All Patients	%/Range
Demographics								
Female, n (%)	42	37.5	70	62.5	—		112	100
Median age at diagnosis, years (range)	64.3	(36.1–73.8)	60.8	(37.9–73.4)	0.0413	*	61.8	(36.1–73.8)
Median age at HDCT/ASCT, years, (range)	64.8	(36.5–74.3)	61.8	(38.2–73.8)	0.0390	*	62.3	(36.5–74.3)
Median body weight at HDCT/ASCT, kg (range)	65.5	(47–109)	80.0	(60–130.5)	0.0001	****	76.0	(47–130)
Median eGFR, mL/min/1.73 m^2^ (range)	94.5	(48–124)	97.1	(34–125)	0.4371	n.s.	96	(34–125)
R-ISS Stage, n (%)								
I	7	16.7	19	27.1	0.2515	n.s.	26	23.2
II	23	54.8	29	41.4	0.1785	n.s.	52	46.4
III	10	23.8	17	24.3	>0.9999	n.s.	27	24.1
No data	2	4.8	5	7.1	0.7095	n.s.	7	6.3
Cytogenetics, n (%)								
High risk	13	31.0	11	15.7	0.0944	n.s.	24	21.4
Standard risk	21	50.0	43	61.4	0.2453	n.s.	64	57.1
No data	8	19.0	16	22.9	0.8125	n.s.	24	21.4
Paraprotein Subtype, n (%)								
IgA	7	16.7	9	12.9	0.5879	n.s.	16	14.3
IgG	28	66.7	49	70	0.8336	n.s.	77	68.8
IgM	0	0	1	1.4	0.4365	n.s.	1	0.9
Light chain only	7	16.7	11	15.7	>0.9999	n.s.	18	16.1
Kappa light chain	25	59.5	48	68.6	0.4131	n.s.	73	65.2
Lambda light chain	17	40.5	22	31.4	0.4131	n.s.	39	34.8

Abbreviations: HDCT, high dose chemotherapy; eGFR: estimated glomerular filtration rate. s denotes significance level: n.s. *p* > 0.05; * *p* ≤ 0.05; **** *p* ≤ 0.0001.

**Table 2 cancers-16-03364-t002:** Treatment overview. Induction, HDCT/ASCT and maintenance.

	Females	%/Range	Males	%/Range	*p*-Value	s	All Patients	%
Induction regimens, n (%)								
Bortezomib/lenalidomide/dexamethasone	29	69	63	90.0	0.0095	**	92	82.1
Daratumumab/bortezomib/lenalidomide/dexamethasone	9	21.4	4	5.7	0.0161	*	13	11.6
Other	4	9.5	3	4.3	0.4218	n.s.	7	6.3
Dartumumab in induction	10	23.8	4	5.7	0.0075	**	14	12.5
Cycles of induction therapy, median (range)	4	(2–4)	4	(2–6)	0.4126	n.s.	4	(2–6)
HDCT/ASCT								
Time to HDCT from first diagnosis, months, median (range)	5.3	(3.3–60.9)	4.6	(2.9–75.3)	0.0056	**	4.8	(2.9–75.3)
HD regimen: treosulfan/melphalan, n (%)	42	100	70	100	>0.9999	n.s.	112	100
CD34+ cells transfused, median (range)	256.7	(98.5–462)	239.8	(90–1369)	0.3814	n.s.	249.9	(90–1369)
Treosulfan AUC (mg*h/L), median (range)	869.9	(578–1583)	830.5	(465–1504)	0.0427	*	833.0	(465–1583)
Treosulfan peak (mg/L), median (range)	343.8	(256–554)	309.0	(168–498)	0.0011	**	330.0	(168–554)
Maintenance therapy, n (%)								
Lenalidomide	39	92.9	65	92.9	>0.9999	n.s.	104	92.9
Daratumumab	0	0	1	1.4	>0.9999	n.s.	1	0.9
Other	0	0	3	4.3	0.2905	n.s.	3	2.7
No maintenance	3	7.1	1	1.4	0.1146	n.s.	4	3.6

Abbreviations: HDCT, high dose chemotherapy; ASCT, autologous stem cell transplantation; eGFR, estimated glomerular filtration rate. s denotes significance level: n.s. *p* > 0.05; * *p* ≤ 0.05; ** *p* ≤ 0.01.

**Table 3 cancers-16-03364-t003:** Outcome of patients after treatments.

	Females	%/Range	Males	%/Range	*p*-Value	s	All Patients	%/Range
Response before HDCT/ASCT, n (%)								
CR	7	16.7	11	15.7	>0.9999	n.s.	18	16.1
VGPR	23	54.8	26	37.1	0.0793	n.s.	49	43.8
PR	5	11.9	24	34.3	0.0133	*	29	25.9
SD	0	0	2	2.9	0.5270	n.s.	2	1.8
No Info	7	16.7	7	10	0.3788	n.s.	14	12.5
VGPR or better	30	71.4	37	52.9	0.0729	n.s.	67	59.8
Survival								
Progression-free survival, months, median (range)	n.r.	(0.25–44.81)	36.7	(2.56–40.53)	0.2644	n.s.	n.r.	(0.25–44.81)
Overall survival, months, median (range)	n.r.	(0.25–44.81)	n.r.	(2.63–42.70)	0.8951	n.s.	n.r.	(0.25–44.81)
Follow up, months, median (range)	—	—	—	—	—		30.95	—
Best response after HDCT/ASCT, n (%)								
sCR	27	64.3	43	61.4	>0.9999	n.s.	70	62.5
CR	12	28.6	12	17.1	0.1629	n.s.	24	21.4
VGPR	2	4.8	7	10	0.4795	n.s.	9	8
PR	0	0	4	5.7	0.2950	n.s.	4	3.6
SD	0	0	1	1.4	>0.9999	n.s.	1	0.9
No Info	1	2.4	3	4.3	>0.9999	n.s.	4	3.6

Abbreviations: HDCT, high dose chemotherapy; ASCT, autologous stem cell transplantation; sCR, stringent complete response; CR, complete response; VGPR, very good partial response; PR, partial response; SD, stable disease; n.r., not reached. s denotes significance level: n.s. *p* > 0.05; * *p* ≤ 0.05.

**Table 4 cancers-16-03364-t004:** Toxicity related to TreoMel HDCT in patients that participated in this study.

	Females	%/Range	Males	%/Range	*p*-Value	s	All Patients	%/Range
Hematologic recovery								
ANC recovery to >0.5 × 10^9^/L, days, median (range)	11	(2–16)	12	(10–26)	0.0119	*	11	(2–26)
Platelet recovery to >100 × 10^9^/L, days, median (range)	12	(7–38)	13	(9–52)	0.2132	n.s.	12	(7–52)
Platelet transfusions, units, median (range)	3	(0–30)	2	(0–16)	0.2348	n.s.	3	(0–30)
RBC transfusions, units, median (range)	1	(0–15)	0	(0–19)	0.0173	*	1	(0–19)
Engraftment syndrome, n (%)	6	14.3	5	7.1	0.3253	n.s.	11	9.8
Hospitalization								
Inpatient stay, days, median (range)	24	(12–89)	21	(16–38)	0.0007	***	22	(12–89)
ICU admission, n (%)	5	11.9	2	2.9	0.1007	n.s.	7	6.3
Infectious complications								
Neutropenic fever, n (%)	41	97.6	68	97.1	>0.9999	n.s.	109	97.3
Neutropenic colitis, n (%)	28	66.7	56	80	0.1223	n.s.	84	75
Bacteriemia, n (%)	17	40.5	23	32.9	0.4240	n.s.	40	35.7
Central line associated blood stream infection, n (%)	3	7.1	18	25.7	0.0732	n.s.	21	18.8
SARS-CoV2 infection, n (%)	4	9.5	2	2.9	0.1949	n.s.	6	5.4
Metabolism								
Parenteral nutrition, n (%)	40	95.2	60	85.7	0.2051	n.s.	100	89.3
Duration of PN, days, median (range)	10	(2–25)	8	(2–20)	0.0025	**		
Refeeding syndrome in patients requiring PN, n (%)	13	32.5	33	55	0.0401	*	46	46
Mucositis, n (%)	19	47.5	37	61.7	0.5585	n.s.	56	50
Acute kidney injury, n (%)	4	9.5	3	4.3	0.4218	n.s.	7	6.3
Hepatic injury, n (%)	2	4.8	3	4.3	>0.9999	n.s.	5	4.5

Abbreviations: HDCT, high dose chemotherapy; ANC, absolute neutrophil count; RBC, red blood cell; ICU, intensive care unit; PN, parenteral nutrition. s denotes significance level: n.s. *p* > 0.05; * *p* ≤ 0.05; ** *p* ≤ 0.01; *** *p* ≤ 0.001

## Data Availability

The data presented in this study are available upon request from the corresponding author.

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
