# Peer review of "Gender-Specific Prognostic Impact of Treosulfan Levels in High-Dose Chemotherapy for Multiple Myeloma"

_cancers, 2024, doi:10.3390/cancers16193364_

Round 1

Reviewer 1 Report

Comments and Suggestions for Authors

The manuscript "Gender-Specific Prognostic Impact of Treosulfan Levels in Hugh-Dose Chemotherapy for Multiple Myeloma" analyzed data from 112 multiple myeloma patients treated with Treosulfan and melphalan and then who had autologous stem cell transplantation for two years. They showed that women had higher interactions with Treosulfan; however, it caused women more side effects compared to men and did not improve overall survival.

The study has an impact in the field, as still few research considers the effects of therapies and their associated impact on females. Their data showed that women responded better with lower levels of Treosulfan. 

I have a few questions for the authors, and if they have this data and include it, I believe it could improve their manuscript. 

Authors showed and mentioned several side effects women have over men when treated with this drug: 

1) Did the treatment with the drug affect women's vs men's complete blood count? Or do they have information on if higher vs lower doses impact their blood count differently?

2) Does it affect blood reconstitution after ASCT? Does the higher or lower dose have any effect on engraftment? 

3) The patients that had poor overall survival did these patients have MM remission, or was it caused by any other hematopoietic malignancy caused by the treatment? It is known that chemotherapy for one condition can trigger gene mutation for another malignancy. 

Overall, the tables were clear and easy to understand, providing the necessary information. The figures were mostly clear. The authors could improve figures 2 and 3, as the fonts and size were too small. They could also maintain the red and blue colors from the first figure.

Author Response

Reviewer 1 report:

The manuscript "Gender-Specific Prognostic Impact of Treosulfan Levels in Hugh-Dose Chemotherapy for Multiple Myeloma" analyzed data from 112 multiple myeloma patients treated with Treosulfan and melphalan and then who had autologous stem cell transplantation for two years. They showed that women had higher interactions with Treosulfan; however, it caused women more side effects compared to men and did not improve overall survival.

The study has an impact in the field, as still few research considers the effects of therapies and their associated impact on females. Their data showed that women responded better with lower levels of Treosulfan.

I have a few questions for the authors, and if they have this data and include it, I believe it could improve their manuscript.

Reply: We thank the reviewer for the helpful comments.

Authors showed and mentioned several side effects women have over men when treated with this drug:

1) Did the treatment with the drug affect women's vs men's complete blood count? Or do they have information on if higher vs lower doses impact their blood count differently?

Reply: We have included this information in Table 4. Male patients had longer time for neutrophil recovery, however, female patients were more likely to require red blood cell transfusions. We have expanded the respective section in the text to illustrate this more clearly. Unfortunately, we do not have information on platelet recovery adjusted for the low/high dose cohorts.

2) Does it affect blood reconstitution after ASCT? Does the higher or lower dose have any effect on engraftment?

Reply: Neutrophil recovery was slower in male patients. We found no difference in the incidence of engraftment syndrome between male and female patients.

3) The patients that had poor overall survival did these patients have MM remission, or was it caused by any other hematopoietic malignancy caused by the treatment? It is known that chemotherapy for one condition can trigger gene mutation for another malignancy.

Reply: We thank the reviewer for this interesting comment. In this present cohort, we found no secondary malignancy so far. Patients are being followed closely after HDCT/ASCT, however, the observed time frame might be too short for secondary hematopoietic malignancies to develop. We have amended the discussion accordingly.

Overall, the tables were clear and easy to understand, providing the necessary information. The figures were mostly clear. The authors could improve figures 2 and 3, as the fonts and size were too small. They could also maintain the red and blue colors from the first figure.

Reply: We thank the reviewer for this comment. As figure 1 shows the comparison between female and male patients and figures 2 and 3 show the comparison between the high and low treosulfan exposition cohorts, we decided not to choose the same colour pallets to avoid confusion. We have increased the size of the graphs and the text to allow them to be read more clearly.

Reviewer 2 Report

Comments and Suggestions for Authors

This is a retrospective study analyzing the effect of gender on the addition of treosulfan to melphalan conditioning prior to ASCT in myeloma patients. The authors present the data in a clear and concise format. Interestingly, they found the higher levels of treosulfan exposure (as evidenced by higher AUC and 30 min peak concentrations) were more likely in female patients and that patients with an AUC > 900 mg*h/l or peak concentration > 400 mg/L had worse OS but similar PFS. Suggesting additional toxicity with this dose.  Notably, the combination of treosulfan and melphalan had significantly higher rates of AEs compared to trials of melphalan alone (i.e. febrile neutropenia in the phase 3 determination trial only occurred in 9% of transplant patients) vs 97% with treosulfan and melphalan. The authors should make this clear rather than the statement that, "As being common after high-dose chemotherapy treatment, nearly all patients developed neutropenic fever" especially since this study is somewhat underpowered to determine gender specific AEs. 

Author Response

Reviewer 2 report:

This is a retrospective study analyzing the effect of gender on the addition of treosulfan to melphalan conditioning prior to ASCT in myeloma patients. The authors present the data in a clear and concise format. Interestingly, they found the higher levels of treosulfan exposure (as evidenced by higher AUC and 30 min peak concentrations) were more likely in female patients and that patients with an AUC > 900 mg*h/l or peak concentration > 400 mg/L had worse OS but similar PFS. Suggesting additional toxicity with this dose. Notably, the combination of treosulfan and melphalan had significantly higher rates of AEs compared to trials of melphalan alone (i.e. febrile neutropenia in the phase 3 determination trial only occurred in 9% of transplant patients) vs 97% with treosulfan and melphalan. The authors should make this clear rather than the statement that, "As being common after high-dose chemotherapy treatment, nearly all patients developed neutropenic fever" especially since this study is somewhat underpowered to determine gender specific AEs.

Reply: We thank the reviewer for this comment. While we agree that the rate of neutropenic fever in our cohort is high, the incidence of febrile neutropenia (FN) reported in the determination trial was exceptionally low. FN Rates typically reported are between 50% and 95% (Kassar et al., Transfusion 2007; Rodriguez et al., BBMT 2020; Yeung et al, JHOP 2020; Martino et al., Ann Hematol 2023), which is in line with the experience at our center. The determination authors do not comment on this issue in their article.

We believe that the question of whether the addition of treosulfan leads to more frequent or more severe side effects is beyond the scope of this current study. It is being investigated as part of the prospective TreoMel trial (NCT05636787) currently conducted at our center.

We have rewritten the respective section in a more neutral manner.